# The Effect of Novel and Environmentally Friendly Foods on Consumer Attitude and Behavior: A Value-Attitude-Behavioral Model

**DOI:** 10.3390/foods11162423

**Published:** 2022-08-12

**Authors:** Chun-Chieh Ma, Hsiao-Ping Chang

**Affiliations:** 1Department of Public Administration and Management, National University of Tainan, No.33, Sec. 2, Shu-Lin St., Tainan 70005, Taiwan; 2Department of Health Industry Technology Management, Chung Shan Medical University, No.110, Sec. 1, Jianguo N. Rd., Taichung City 40201, Taiwan; 3Department of Medical Management, Chung Shan Medical University Hospital, No. 110, Sec. 1, Jianguo N. Rd., Taichung City 40201, Taiwan

**Keywords:** plant-based meat alternatives, novel and environmentally friendly foods, value-attitude-behavior model, perception of green value, animal welfare value, product knowledge

## Abstract

Extreme weather conditions have intensified due to manufactured environmental damage in recent years. To reduce the environmental impact on the Earth, many consumers seek to change their dietary patterns to protect the environment and voluntarily switch to a vegetarian diet. Past studies have found that the transition from nonvegetarian to vegetarian is not easy, but promoting the consumption of alternative foods such as plant-based meat alternatives should help consumers gradually reduce their dependence on meat during the transition period of changing their eating habits. This study was designed to apply the value-attitude-behavior model (VAB) to study the consumption attitude and behavior of novel and environmentally friendly foods such as plant-based meat alternatives, and the novelty of plant-based meat alternatives was included as an intervening variable for discussion. In this study, 376 valid questionnaires were collected from college students in Taiwan, and the recovery rate of valid questionnaires was 94%. It was found from the analysis of results that perceptions of green value and animal welfare value had a significantly positive effect on attitude, while attitude and product knowledge also had a significant positive effect on behavior; however, the novelty of plant-based meat alternatives products did not have an interference effect on the relationship between product knowledge and behavior. Based on the research findings of this study, it is suggested that when introducing plant-based meat alternatives products, food companies should not only let consumers understand that they are based on environmental friendliness and animal welfare values but also enhance the marketing and promotion of product knowledge to increase consumers’ confidence in purchasing plant-based meat alternatives and reduce their consumption concerns.

## 1. Introduction

As global warming gradually worsens, improving the environment has become an urgent issue for all consumers. Being one of the leading causes of global warming, 80% of greenhouse gases produced by agriculture come from animal husbandry [1]. If the current diet structure of human beings is maintained until 2050, the emissions of greenhouse gases and the use of water resources, land, and fertilizers are likely to exceed the limit that the Earth can handle [2]. The greenhouse gases released from food production account for about 20% to 30% of global greenhouse gas emissions [3], and if meat is completely eliminated from the diet, it will reduce greenhouse gas emissions by about one third [4].

According to Pimentel and Pimentel [5], consumers can change their dietary habits and consumption behaviors to achieve the effects of environmental improvement. In this regard, the alternative food production process, plant-based meat alternatives, consumes 46% less energy, produces 90% less greenhouse gas emissions, uses 93% less land and 99.9% less water than the production of beef [6]. Plant-based meat alternatives are a novel alternative food that meat eaters should consume daily. It can effectively improve the environment as a novel alternative food. If consumers can reduce meat consumption and gradually switch to plant-based meat alternatives substitutes, it could effectively improve global warming.

“Plant-based meat alternatives substitutes” is a reasonably general term; for this study, the term should include analogs that closely approximate whole-muscle animal meat in texture, flavor, and appearance, as well as reconstituted products that mimic processed meat, such as burgers, patties, sausages, and chicken nuggets. Meat substitutes can be classified as plant-based (soy, pea, gluten, etc.), cellular (in vitro or cultured meat), and fermented (fungal protein) since plant-based protein can be used directly to build meat-like substitutes [7]. As such, plant-based meat alternatives substitutes, substitutes, or substitutes represent a significant sector of this emerging and rapidly growing industry. These non-traditional foods attract investment, research, consumer curiosity, and media attention. Over the past few years, global food scientists’ reinvestment in alternative research has increased research publications on meat alternatives. As “alternatives” to traditional animal-derived foods, they are often promoted as “healthier” (than meat) and sustainable new foods [8].

The meat alternatives consumers showed the most probable willingness to purchase were found to be plant-based proteins. Plant-based meat alternatives tend to have a wider range of sources than other alternative protein sources [9]. Those willing to buy alternatives to plant-based meat mainly chose this option because it tended to be more widely available on the market [10]. However, not all animal-food alternatives are sustainable; some are even ultra-processed [11]. Furthermore, in addition to technical barriers to mimicking meat texture and flavor, other potential barriers to food safety and nutrition have not yet been adequately addressed [7,12]. For these reasons, some problems still need to be overcome, including technological, sensory, nutritional, health, and safety challenges in the development of the alternative meat market [13].

Currently, the development of plant-based meat alternative products in Taiwan is in its infancy. However, the price of local plant-based meat alternatives products in Taiwan is high, and the marketing and promotion of plant-based meat alternatives are inadequate. Therefore, it has become critical to identify the key factors that can attract consumers to buy alternatives to plant-based meat. However, few studies have been conducted on the consumer behavior of plant-based meat alternatives, a new type of vegetarian meat. Additionally, the limited research conducted on green foods has tended to use the Theory of Planned Behavior for research and discussion, but little research has been conducted on evaluating consumers’ perception of green value by their perceptions of its green features and price. Finally, research on whether consumers’ perceived animal welfare value affects their consumption attitudes and purchase behaviors towards plant-based meat alternatives is also limited. This study aimed to conduct a behavioral study on consumer values and attitudes toward plant-based meat products in Taiwan.

Based on the value-attitude-behavior (VAB) model, this study deconstructs the characteristics of the key factors that influence consumers’ purchase behavior. Furthermore, Torri et al. [14] noted that younger consumers are more receptive to new foods. Therefore, this study was designed to investigate the key factors that influence the purchasing behavior of plant-based meat alternatives using the VAB model with college students in Taiwan as primary research subjects. Secondly, it investigated whether the novelty of such plant-based meat alternative products can act as an intervening variable to affect consumers’ subjective judgment of new food products. Finally, the relationship between product knowledge and behavior was explored regarding the amount of knowledge they had about the product. As a result of the analysis, this study not only deconstructed the key factors that influence consumers’ attitudes and purchase intentions toward plant-based meat alternatives but also suggested marketing strategies for the future introduction of new products such as plant-based meat alternatives. This study is expected to provide suggestions for the development of alternative food industries, which will lead to a change in dietary patterns to achieve the ultimate goal of environmental protection.

## 2. Literature Review

### 2.1. Alternatives to Meat

In recent years, plant-based meat alternatives, alternatives to meat, have attracted a lot of attention in academia and social media. Plant-based meat alternatives can be defined as plant-based foods with vegetable protein, without animal ingredients, and processed to have a taste and appearance similar to that of meat [6]. In the past, such products were considered to be developed for vegetarians, but more and more products are being developed for non-vegetarian-oriented customers.

Groups of consumers are interested in plant-based meat alternatives products. The willingness to replace or reduce meat consumption can depend to a large extent on current dietary habits. To date, research suggests that the willingness to try plant-based meat alternatives is high compared to other alternatives, but the proportion of people who frequently consume meat substitutes is low within the population, according to the definitions of the Smart Protein survey [15]. Flexitarian individuals consume meat, but intend to reduce their meat intake and consume a higher share of plant-based foods. Omnivores consume meat frequently, and their diet includes all food groups. Pescatarians consume seafood, but not other types of meat. Vegetarians do not consume meat but do consume other animal-based products, such as eggs or dairy. Finally, vegans do not consume animal-based products [16].

Proponents of plant-based meat alternatives see it as a means to reduce animal agriculture and to contribute to environmental sustainability, as animal agriculture is one of the industries that uses the most land and water resources and is a significant source of greenhouse gas emissions [5,17,18]. How to develop and sell alternatives products in the future is an essential topic for researchers and the food industry. It has been noted that for some consumers, the choice to purchase plant-based meat alternatives is a philosophically oriented self-identification [19], but for others, plant-based meat alternatives are an unnatural and even disgusting product [20].

Past research has suggested that plant-based meat alternatives can significantly reduce the proportion of consumers who buy meat from livestock [21], so many manufacturers are working to create a plant-based hamburger steak that tastes and looks exactly like a regular beef hamburger steak. Regardless of the type of plant-based meat alternative product, the ultimate goal is to replace the current consumer demand for meat products to achieve the most significant benefit in improving the environment [22].

However, not all animal-food substitutes are sustainable; some are even ultra-processed. In addition, there are concerns about safety and labeling, and consumers demand clear information and regulation. Challenges in this field are connected with food design and technology, sensory science, nutrition, and dietetics. Furthermore, good selections and combinations of foods are essential to achieve consumer acceptance while preventing nutritional deficiencies in those who choose this diet [11,23].

Aside from technological hurdles on mimicking meat texture and flavor, food safety and nutrition present other potential obstacles that have not been adequately addressed. For example, the inclusion of widely available yet allergenic plant proteins, the addition of a large variety of ingredients and additives to create sensory characteristics, the potential adverse chemical changes for heat-sensitive compounds, and possible microbiological contamination must be systematically investigated [12,24]. For these reasons, despite the notable initial success, there are challenges ahead for plant-based meat alternatives.

### 2.2. Value-Attitude-Behavior Model

Rokeach [25] defined value as a long-lasting belief, a specific pattern of behavior, and social cognition that can promote the rapid adaptation to the environment [26], while attitude is the overall assessment of a product or service by consumers [27], and positive attitudes are formed when people are attracted to a product [28]. Ajzen and Fishbein [29] pointed out that attitude is an essential antecedent to determining behaviors, while behavior is a consumer’s action influenced by his or her attitude. The study of Engel et al. [27] showed that consumers’ attitudes towards a product or service affect their behavior, such as purchasing behavior or willingness to pay a premium. Attitudes can be used to predict people’s environmental behavior [30]. The better consumers’ attitudes towards the environment, the less likely they are to purchase products that pollute the environment [31].

Past studies have shown that values are the most critical structure to establish attitudes and behaviors [32], in the order of value-attitude-behavior. The VAB (value-attitude-behavior) model has been shown to help explore consumers’ behavior in purchasing organic food [33] and in explaining consumers’ environmental behavior [34].

### 2.3. Hypotheses

#### 2.3.1. Perception of Green Value and Attitude

Perceived value affects customers’ attitudes towards the target object [35]. For business operators, outstanding value can distinguish the product from the competition in the market [36]. Perceived value is a fundamental reason consumers keep buying and a key factor that influences consumers’ purchase intention [37]. Since perceived value significantly impacts the business performance of companies, it can also be used to increase consumers’ purchase intention by assigning specific values to products [38,39]. Due to the trend of sustainable development, the study of perceived value tends to explore the theme of environmental protection, and thus the concept of perception of green value has been developed [40].

The perception of green value is the most critical value for environmentally conscious consumers [41]. Specifically, perception of green value is the subjective assessment of consumers’ concerns, expectations, and needs for green development. In other words, it is the consumer’s general assessment of the benefits of a product or service based on their concern for the environment, their expectation of sustainable development, and their desire for green development [42]. Since the functional and emotional values of the perception of green value can be used to predict consumer attitudes towards green products [43], accordingly, if the perception of the green value of a product can meet consumer expectations, it can lead to positive attitudes and purchase behavior [44].

In terms of consumers’ attitudes towards plant-based meat alternatives, the belief that people surrounding the individual are reducing their meat consumption should activate their moral obligation to follow in the same direction [45]. Furthermore, research shows that the stronger the perception that significant others approve of one’s reduction in meat consumption, the greater the individual’s awareness of the consequences of this behavior [46]. Furthermore, people care whether their behavior is moral to others and are motivated to maintain a positive moral self-image [47] and to belong to a moral group [48]. Based on the above study, it was deduced that:

**H1.** 
*Perception of green value positively and significantly influences consumers’ attitudes towards plant-based meat alternatives.*


#### 2.3.2. Animal Welfare Value and Attitude

As the global population grows and meat consumption increases, alternative food production strategies should be adopted to address environmental and animal welfare issues [2]. Webster [49] noted that consumers’ motivations for buying organic food include animal welfare and green consumption, as animal welfare occupies an important place in the ethical code of the consumer society. Therefore, the criteria for how to properly treat animals are not only determined by the breeder but are also constrained by the animal welfare values of the consumer society [50].

Since animal welfare is a moral issue [51], people’s attitudes towards it can vary depending on their culture, area of residence, time, or personal factors [52,53]. Studies in the related literature pointed out that vegetarians based on animal welfare believe that there is a solid emotional bond between humans and animals [54]. Some people have an attitude of rejecting animal products based on animal welfare or health factors [55] and believe that animals have the same perceptive capacity as consumers and should be treated with animal welfare value [56]. Based on those mentioned above, people are motivated to reduce their meat consumption for different reasons, for example, animal welfare, environmental, and health concerns [45,57]. These motivations are not mutually exclusive; however, it is possible to identify a trend where animal-rights and ecological concerns are more likely to be found in those who completely exclude meat from their diet, whereas less morally relevant reasons, such as health concerns, seem to mainly motivate those who deliberately choose only to reduce meat consumption [58,59]. From the above studies, the following hypothesis was deduced:

**H2.** 
*Animal welfare value positively and significantly influences consumers’ attitudes towards plant-based meat alternatives.*


#### 2.3.3. Consumers’ Attitude and Purchase Behavior

Attitude is an essential predictor of consumer behavior towards energy conservation and environmental protection [60,61], and consumer attitudes towards environmental issues and eco-social benefits affect their green purchase behavior [62]. If observed at the level of consumer attitudes and environmental behaviors, those who have a positive attitude towards environmental behaviors are more likely to engage in environmental behaviors such as recycling resources, taking public transport in a sustainable and environmentally friendly way, joining environmental organizations, saving energy, and using green electricity [63,64,65,66].

Continuing from the above, attitude and perceived behavioral control positively influence behavioral intentions [67,68]. A positive and significant relationship was observed between environmental attitude and energy saving behavior in previous studies [69], environmental attitude positively influenced the willingness to pay for environmental activities [70], and consumer attitude influenced their purchase of natural products [33,71], which indicated that measuring the attitude of student groups was effective in predicting the main reasons for purchasing organic food. Related studies also indicate that consumer concerns impact their purchase intentions [72]. Accordingly, environmental issues and knowledge are among the most influential factors in green buying intentions among young consumers. On the basis of the above, this study deduced that:

**H3.** 
*Consumers’ attitude towards plant-based meat alternatives positively and significantly influences their purchase behavior of plant-based meat alternatives.*


#### 2.3.4. Product Knowledge and Purchase Behavior

Consumer product knowledge refers to their experience with the product or their accumulated knowledge about the product [73], and if consumers do not know anything about the product, it is difficult for them to link to the product and make a positive evaluation [74]. The primary sources of product knowledge are experience with the product and critical advertising messages that can influence consumers’ choice of the product [75]. Product knowledge is considered an internal variable that influences consumers’ evaluation of a product. However, it may influence consumers’ purchase intention or actual consumption behavior [76,77,78]. Hines et al. [79] argue that knowledge affects consumers’ intentions and behavior.

As mentioned above, knowledge is a crucial factor influencing consumer behavior, and consumers can measure the value of a product and the risk they need to take when purchasing it through knowledge [80]. In this regard, there are significant positive relationships between product knowledge, attitudes, and behaviors [81], and product knowledge positively and significantly affects purchase intentions [82]. The more knowledge consumers have, the more they value the features of high-quality products and the more they are willing to pay for them [83,84]. Compared to the past, consumers of green products would like to receive more detailed information and knowledge of the product to evaluate the business operators to green development [40,85]. On the basis of the above, this study deduced that:

**H4.** 
*Consumers’ product knowledge positively and significantly influences the purchase behavior of plant-based meat alternatives.*


#### 2.3.5. Novelty of Plant-Based Meat Alternatives, Product Knowledge, and Purchase Behaviors

Many factors affect the choice of food, among which novelty is one of the critical factors affecting people’s attitudes towards food [86]. Fundamentally, Lin and Huang [87] found in their study that psychological benefits, a strong desire for knowledge, and the pursuit of novelty are the main motivators for consumers to choose green products. Furthermore, due to globalization, people worldwide are exposed to more and more foreign food cultures, and multicultural cooking methods are changing the food concepts that many people have had in the past [88]. Related studies have also shown that consumers are becoming more receptive to new foods due to the growing global population [89].

As mentioned above, consumers’ purchase intentions are influenced by their perceptions of products and their pursuit of novelty, and consumers seek novelty in their food purchases [90]. However, relevant studies also pointed out that the lack of understanding in new food processing technologies may be one of the reasons why nonprofessionals oppose novel foods. Therefore, if consumers are informed about the benefits of new foods (e.g., health benefits and reduced environmental impact), they may be more likely to buy these products [91]. On the basis of the above, this study deduced that:

**H5.** 
*The more substantial the novelty of plant-based meat alternative food products, the more it will affect the relationship between consumer product knowledge and plant-based meat alternatives purchase behavior.*


## 3. Methodology

### 3.1. Sample and Data Collection

The research participants in this study were students in the department of food science or nutrition of Taiwan University. This study used a paper questionnaire and adopted purposive sampling to conduct a research investigation. The questionnaire distribution started in March 2021 for two months. A total of 400 questionnaires were collected and 24 invalid, incomplete, and random responses were deducted. There were 376 valid questionnaires returned and the effective questionnaire recovery rate was 94%. Table 1 shows the analysis of the sociodemographic sample structure. Most of the subjects were female (*N* = 273, 72.6%), and there were 376 subjects. In terms of monthly disposable amount, the highest percentage of subjects had the amount from New Taiwan dollars (NTD) 5000 to NTD 9999 (*N* = 183, 48.7%), followed by those below NTD 4999 (N = 82, 21.8%), and the lowest number of subjects had the amount of NTD 10,000 to NTD 14,999 (*N* = 75, 19.9%). There were 122 subjects who had purchased alternatives to plant-based meat (*N* = 122, 32.4%), and 254 who had not (*N* = 254, 67.6%). Regarding whether they paid attention to environmental issues, 317 subjects paid attention (*N* = 317, 84.3%) and 59 subjects did not (*N* = 59, 15.7%), representing 15.7%.

### 3.2. Data Analysis Procedures

This research adopted SPSS version 22.0 (Statistical Package for Social Sciences) (IBM Corp.: New York, NY, USA) and AMOS (Analysis of Moment Structure) version 22.0 (IBM Corp.: New York, NY, USA) to analyze the data. The data analysis steps in this study consisted of seven steps. Step 1 is narrative statistical analysis, Step 2 is the reliability analysis, Step 3 is the confirmatory factor analysis, Step 4 is Pearson’s correlation analysis, Step 5 is the structural equation modeling (SEM), Step 6 is the fit analysis of hypothetical structural models, and Step 7 is the hierarchical regression analysis.

### 3.3. Questionnaire Design

Based on the literature review, this study will use the VAB model to explore consumers’ attitudes toward plant-based meat alternatives products in terms of perception of green value and animal welfare value and add the novelty of plant-based meat alternatives as an interference variable to explore its impact on the relationship between product knowledge and purchasing behavior. Figure 1 presents the research framework for investigating the relationships among perception of green value, animal welfare value, attitude, knowledge of the product and purchase intention. The four-item scale in the section on the perception of green value of the questionnaire included items such as “I think plant meat is an environmentally friendly product” and was adapted from Williams and Soutar with a Cronbach’s α of 0.868 [92]. The three-item scale in the animal welfare value section of this survey included items such as “I don’t think eating meat from livestock raised animals is in line with animal welfare, because their lives deserve to be respected” and was adapted from Zhuang and Liu with a Cronbach’s α of 0.870 [93]. The two-item scale in the attitude section of this survey included items such as “I think it is right to buy plant-based meat products for environmental sustainability” and was adapted from Tsen et al. with a Cronbach’s α of 0.879 [94]. The three-item scale in the purchase intention section included items such as “I would like to buy plant-based meat alternatives that promote environmental sustainability” and was adapted from Follows and Jobber with a Cronbach’s α of 0.910 [95]. The four-item scale in the product knowledge section of the questionnaire included items such as “Information about plant-based meat alternatives influences my intention to purchase” and was adapted from Brucks with a Cronbach’s α of 0.910 [96]. The two-item scale in the attitude section of this survey included items such as “I like to try the novelty of plant-based meat alternatives products” and was adapted from Pliner and Hobden with a Cronbach’s α of 0.910 [97]. The Cronbach’s α coefficients should be at least 0.50 and preferably greater than 0.70 [98], and on each scale, the questionnaire is measured on a 5-point Likert scale ranging from 1 (strongly disagree) to 5 (strongly agree). Details about the source of the questionnaire items are highlighted in Table 2.

## 4. Results

### 4.1. Reliability and Validity

The primary function of the reliability analysis is to test whether the results of each variable measurement are stable and consistent, and to what extent. If the value of the reliability coefficient is higher, the internal reliability of this measurement is consistent and reliable [99]. The Cronbach’s alpha value should be at least greater than 0.50, preferably greater than 0.70 [91]. The Cronbach’s α of each construct in this study was more significant than 0.7 (see Table 3 for details), indicating that the measurement tool had considerable reliability. That is, all the measurement elements of the six constructs of the scale had internal consistency, and a high degree of stability of the questionnaire was maintained. The higher the value of composite reliability (CR), the higher the proportion of the actual variance to total variance, that is, the higher the internal consistency, which can be considered as the internal consistency of a construct. The composite reliability value (CR) should be higher than 0.6 [100]. The CR of the variables in this study ranged from 0.781 to 0.912, indicating that this model had good internal consistency. The extracted average variance (AVE) is the degree to which all the variables measured in the latent variables can explain the latent variables. The higher the AVE, the higher the degree of the latent variables that are explained by the measured variables. The AVE for each factor was between 0.487 and 0.784, which is higher than the recommended reference of 0.36 [101]. However, the factor loading value (0.53–0.92) was higher than the recommended level of 0.5 [102]. Means, standard deviations, and correlations among the constructs are presented in Table 4.

Both perception of green value and animal welfare value had a significant positive correlation with attitude (γ = 0.591, γ = 0.458, *p* < 0.01). That is, when consumers thought that plant-based meat alternatives had a higher perception of green value or animal welfare value, their attitude towards plant-based meat alternatives also became more positive. There was also a significant positive correlation between attitude and behavior (γ = 0.699, *p* < 0.01), that is, when consumers had more positive attitudes towards plant-based meat alternatives, the purchase behavior of plant meat also increased. There was a significant positive correlation between product knowledge and behavior (γ = 0.571, *p* < 0.01). In other words, when consumers believed they had more product knowledge about plant-based meat alternatives, their purchasing behavior for plant-based meat alternatives also increased. There was a significant positive correlation between the novelty of plant-based meat alternatives products and the perception of green value and animal welfare value (γ = 0.440, γ = 0.301, *p* < 0.01). If consumers were more receptive to the novelty of plant-based meat alternative food, their perception of green value and animal welfare value increased too. There was a significant positive correlation between the novelty of plant-based meat alternatives and attitude (γ = 0.471, *p* < 0.01). This means that if consumers were more receptive to the novelty of the food of plant-based meat alternatives, their attitudes toward it would be more positive. There was a significant positive correlation between the novelty of plant-based meat alternatives food and behavior (γ = 0.580, *p* < 0.01). If consumers were more receptive to the novelty of plant-based meat alternatives products, the purchase behavior of plant-based meat alternatives would also increase.

### 4.2. Structural Equation Modeling and Empirical Analysis

This research applied SEM using AMOS 22.0 to assess the path relationships among the perception of green value, the value of animal welfare, attitude, purchase intention, and knowledge of the product. The results indicated that the measurement model provided a good fit for the data (χ^2^/df = 2.351, GFI = 0.930, AGFI = 0.900, CFI = 0.964, NFI = 0.940, SRMR = 0.049, RMSEA = 0.060). The χ^2^/df ratio is below the value of 3 (Carmines, 1981), the GFI, AGFI, CFI NFI, and IFI exceeded the recommended threshold of 0.90 [96,97,98,99,100,101,102,103,104], and the values of RMSEA and SRMR were below the cutoff value of 0.08 [105]. This indicated that the approach used in this study to model the examined data was appropriate. The hypotheses that test the results of the model data are provided in Figure 2 and Table 5.

The path coefficient of perception of green value and attitude has reached a significant level (γ_11_ = 0.769, *p* < 0.001), indicating that perception of green value and attitude were positively and significantly correlated, meaning that the higher the consumers’ perception of green value, the higher their attitude towards plant-based meat alternatives. So, H1 was supported. The path coefficient between animal welfare value and attitude reached a significant level (γ_12_ = 0.246, *p* < 0.001), indicating that animal welfare value and attitude were positively and significantly correlated, i.e., the higher the consumers’ animal welfare value of consumers, the higher their attitude towards plant-based meat alternatives. Therefore, H2 was supported by empirical data. The path coefficient of attitude and behavior reached a significant level (β_12_ = 0.719, *p* < 0.001), indicating a positive and significant correlation between attitude and behavior, which means that the higher the attitude towards plant-based meat alternatives, the higher the purchase behavior of plant-based meat alternatives. Therefore, H3 was supported by empirical data. The path coefficient of knowledge and behavior reached a significant level (γ_23_ = 0.450, *p* < 0.001), indicating that knowledge and behavior were positively and significantly correlated, showing that the more knowledge consumers have about plant-based meat alternatives products, the higher the purchase behavior of plant-based meat alternatives. Therefore, H4 was supported.

### 4.3. Testing Interference Effects

Hierarchical regression was used to analyze the interference effect of the novelty of plant-based meat alternatives on the relationship between knowledge of the product and purchase intention. The analysis results are presented in Table 6. The regression model M1 in this table shows the impact of knowledge of the product on purchase intention in the first stage, with the value β = 0.571 and *p* < 0.001. The coefficient reached a significant level and presented a significant positive correlation. The regression model M2 shows the influence of the novelty of plant-based meat alternatives on the intention to purchase in the second stage, with its value β = 0.393 and *p* < 0.001. The coefficient reached a significant level and showed a positive and significant correlation. The regression model M3 shows that after adding the product term of product knowledge x novelty of plant-based meat alternatives in the third stage, the variance explained (R^2^) did not change, the change value of the variance explained (ΔR^2^) was 0.000, and its value was β = 0.014, *p* > 0.05. The coefficient did not reach a significant level, indicating that the novelty of plant-based meat did not interfere with the relationship between “product knowledge and behavior”. From the regression model M3 in Table 6, it can be seen that by adding “if the average number of the novelty of plant-based meat alternatives products was taken as the cut-off point”, the samples were divided into two groups, with high-novelty of plant-based meat alternatives food and low-novelty of plant-based meat alternatives food for separate analysis; the line segment of the regression model of product knowledge and purchase intention drawn was as shown in Figure 3. Whether consumers’ preference for the novelty of plant-based meat alternatives was high or low, the effect between product knowledge and purchase intention could not be increased. Therefore, Hypothesis H5 was not supported.

## 5. Conclusions and Suggestions

### 5.1. Conclusions

#### 5.1.1. Relationship between Perception of Green Value and Attitude

Energy consumption in the production of plant-based meat alternatives is much less than that in traditional meat production, which is desirable for consumers who care about the Earth and the environment [22]. Although currently in Taiwan, the prices of plant-based meat alternatives products that can be purchased on the market are relatively high, there are still consumers willing to buy plant-based meat alternatives for the sustainable development of the environment. According to this research result, the perception of green value positively and significantly affected the attitude towards plant-based meat alternatives, which means that when consumers felt a higher perception of the green value of plant-based meat alternatives, their attitude towards plant-based meat alternatives would also be higher. This result is the same as the research findings of [44]. That is, if the perception of green value can meet the expectations of consumers, it can prompt them to have a positive attitude. From the implications of perspective of the management practice, it means that consumers’ perception of green value will affect their attitude toward green products. Specifically, when the benefits a product can provide to the environment are more in line with consumer expectations, consumers are more likely to have a positive attitude towards the product. Due to the characteristics of the plant-based meat alternatives that can reduce resource consumption and promote environmental friendliness [5,17,18], it is suggested that the government can consider incorporating such green products into environmental protection policies or environmental education to improve the public’s awareness of environmental protection products and encourage the integration of environmental protection behaviors into daily life, to achieve the effect of improving the environment, thus improving the quality of green products.

#### 5.1.2. Relationship between Animal Welfare Value and Attitude

The results of Webster [49] showed that consumers’ motivations to buy green products include animal welfare and green consumption. The results of this study also indicate that animal welfare value would positively and significantly affect consumers’ attitudes towards plant-based meat alternatives. This means that when consumers felt the higher animal welfare value of plant-based meat alternatives, their attitude toward plant-based meat alternatives was also higher. This result should have practical implications for management. That is, the animal welfare value felt by consumers will affect their attitude towards green products. When a product’s ability to improve animal welfare is more in line with consumer expectations, consumers are more likely to have a positive attitude towards the product; relatively, producers of plant-based meat alternatives are more able to use animal welfare value to predict consumers’ attitude towards their products. The plant-based meat alternative production process does not harm animals, which can effectively improve consumer acceptance of products [2,54]. As a result, business operators must advertise products from an animal-friendly perspective to help differentiate the market from other meat products.

#### 5.1.3. Relationship between Attitude and Behavior

This study showed that consumers’ attitudes toward plant-based meat alternatives positively and significantly affected consumers’ purchase behavior, which means that consumers who had a positive attitude towards plant-based meat alternatives would be more willing to buy plant-based meat alternatives. The research findings are similar to those of Homer and Kahle [33]. That is, the attitude of consumers affected their behavior to buy natural products. It is also similar to the research results of Yazdanpanah and Forouzani [71]. That is, attitude was the main predictor of the purchase of organic food, and the intention of young consumers’ intention to buy green products could be influenced by attitude. Regarding the implications of management practices, consumers’ attitude towards green products can affect their purchase behavior; the more positive the attitude, the more likely they will have purchase behavior and continue to buy or be willing to pay a premium. Therefore, business operators can engage from various perspectives such as green, environmental protection, perception, animal welfare, animal protection, and morality, to stimulate the positive attitude of consumers and make them willing to continue to buy and pay higher prices.

#### 5.1.4. Relationship between Product Knowledge and Behavior

It was found from the results of this study that product knowledge could positively and significantly affect consumers’ purchase behavior of plant-based meat alternatives, which means that when consumers thought that they had more product knowledge about plant-based meat alternatives products, their purchase behavior of plant-based meat alternatives would increase accordingly. The results of this study are the same as those of Al-Shabib et al. [81], which means a significant positive correlation between knowledge, attitudes, and behaviors of the product. They are also the same as the results of Lin and Chen [82], that is, the knowledge of the product positively and significantly affected the intention to purchase. This finding of research in management practice implies that the subjective perception of the amount of knowledge they have about green products will affect their purchase behavior. The more knowledge consumers have, the more likely it is to generate purchase behaviors and make them continue to buy or be more willing to pay a premium. Therefore, business operators can strengthen the disclosure of the product and increase the information channels of green products through which consumers strengthen their purchase behaviors.

#### 5.1.5. Interference Effect of Novelty of Plant-Based Meat Alternatives on Product Knowledge and Behavior

The study by Coderoni and Perito [91] found that increasing consumer knowledge of the product about the novelty of plant-based meat alternatives foods may make them more willing to purchase these products. However, this is different from the findings of this study. The results of this study indicate that the novelty factor of alternative plant-based meat foods failed to interfere with the relationship between product knowledge and behavior. This means that regardless of how novel the product was, the degree of acceptance of the novelty of plant-based meat alternative foods could not impact the relationship between product knowledge and behavior. However, it is also possible that the research participants were college students who want to try out new foods. It is speculated that the possible reason is that the influence of product knowledge on Taiwanese consumer behavior is strong, and the research participants are mainly young college students attracted to the novelty of plant-based meat alternatives foods. The implication of management practice on research results is that the novelty of the product may not have an impact on the relationship between product knowledge and behavior, which means that the novelty of plant-based meat alternative products may not be one of the critical factors that drives consumers to make decisions about purchasing environmentally friendly products. Although plant-based meat alternatives are novel foods, for Taiwanese consumers, fully understanding the information about the product is the critical factor affecting their intention to buy. Given this, it is recommended that when manufacturers promote their plant-based meat alternatives products, they should focus on increasing consumers’ awareness and understanding of products effectively. When new elements are used in products (such as the use of new processes, new materials, and new forms), they should consider the sense of fear caused by unfamiliarity to consumers and try to use simple and easy-to-understand advertising methods to allow consumers to receive product information to increase their product knowledge effectively, thus reducing consumers’ fear of the novelty of plant-based meat alternative products to increase consumers’ attitude toward and acceptance of novelty products.

### 5.2. Suggestions

This study showed that perception of green value and animal welfare value were key factors affecting consumers’ attitudes towards plant-based meat alternatives products. Therefore, the following recommendations were proposed by this study.

First, perception of green value refers to the perception and evaluation based on their efforts and rewards when purchasing green products or services, and consumers have expectations for the ability of green products to promote sustainable development [17,22]. Therefore, it is suggested that when consumers buy future products related to plant-based meat alternatives, they should prioritize whether they can improve environmental problems. Second, in traditional animal husbandry, no matter how perfect the humane slaughter procedures are, it is still inevitable that the lives of animals are taken. Alternatives to plant-based meat can effectively improve the above situation [50]. Since the raw materials of plant-based meat alternatives do not contain animal ingredients, if plant-based meat alternatives can be used to replace the demand for meat, it would significantly improve the welfare for livestock animals, and this value becomes an essential factor in affecting consumers’ attitude towards plant-based meat alternatives.

Third, given the findings of this study, consumers’ attitudes towards plant-based meat alternatives could positively and significantly affect consumers’ purchase behavior, which is also consistent with the research results of Sapci and Considine [69], Yazdanpanah and Forouzani [71], Yadav and Pathak [72], and others. That is, if consumers can have a positive attitude towards green products, it can promote their purchase behavior at the same time. In this regard, it is suggested that plant-based meat-alternative producers should be able to strengthen the perception of green value and animal welfare value of plant-based meat alternatives through marketing and propaganda activities combined with educational activities for consumers to have a positive attitude towards plant-based meat alternatives, to stimulate consumers to purchase plant-based meat alternatives more effectively. Fourth, since product knowledge can positively impact consumer behavior, if consumers can be provided with more knowledge about green products, it can promote their behavior of purchasing the products. For this, it is suggested that business operators should be able to increase consumer product knowledge through effective advertising methods, and the government should also incorporate green products such as plant-based meat alternatives into environmental protection policies. At the same time, various channels, such as advocacy for food safety and food agriculture education, can also be used to better understand alternative food or green food.

Finally, the results of this study indicate that the novelty of plant-based meat alternative products did not affect the relationship between product knowledge and behaviors; this is probably because the novelty of the product decreased as consumers’ product knowledge of consumers increased. This may be the cause of a decrease in novelty of a product when consumers become familiar with the various characteristics. Currently, most of the plant-based meat alternative products sold in Taiwan are imported, resulting in a high degree of similarity to the product and low novelty. In this regard, if Taiwanese companies can develop their plant-based meat alternative products with local characteristics, they may be able to effectively enhance the novelty of such green products, realize the innovation of green products, and promote sustainable development simultaneously.

## 6. Contributions and Limitations

### 6.1. Contributions

(1) Research Using VAB Model

Although the theory of planned behavior (TPB) has been used extensively to study consumer behavior in the past, this study chose to adopt the VAB model, which can be used to investigate the characteristics of different products and services through different value variables. The results of this study showed that this model could be used to investigate the value, attitudes, and behaviors of consumers toward green foods such as plant-based meat alternatives.

(2) Research on Perception of Green Value and Animal Welfare Value

There are many studies on green perceived value and animal welfare value, but there are few studies on novelty food using these two values simultaneously. This study collated the unique values of plant-based meat alternatives products and further explored them using the VAB model. It was found that both values could positively and significantly affect consumer attitudes toward plant-based meat alternatives, and the findings of this study will be helpful for future studies by future researchers.

(3) Research with Novelty of Plant-Based Meat Alternatives as an Intervening Variable

There have been many studies on the novelty of foods in the past, but very few studies have used it as an intervening variable. In this study, the novelty of plant-based meat alternatives was used as an intervening variable, and it was found that the relationship between knowledge of the product and behavior was not influenced by the moderation of the novelty of plant-based meat alternatives. This means that the relationship between product knowledge and behavior was not affected regardless of the high or low novelty of the food. This research finding will help future researchers in their research development.

### 6.2. Limitations

Regarding research limitations, first of all, this study faced the limitation of sampling the research participants. For Taiwanese consumers, the novelty of plant-based meat alternatives is not familiar to university students, so this study tried to choose the department of food science or nutrition of Taiwan universities as the research sample. According to data from the Taiwan Ministry of Education for the 2022–2022 school year, the ratio of male to female students in food science or nutrition is 25–28% male and 72–75% female, which is similar to the percentage of 27.4% to 72.6% in the research sample of this study. Only students in the Department of Food Science or nutrition at Taiwan University were taken as research participants, resulting in a limited sampling scope so that the results of this study cannot be extrapolated to consumers of all age groups. Additionally, the participants have backgrounds in related fields such as food science, environmental science, and nutrition. Your experience will influence their different thoughts or acceptance of your intention to buy plant-based meat alternatives and the study results. Secondly, the study did not analyze demographic attributes (such as age and the amount of disposable monthly). Plant-based meat alternatives currently on the market are novel foods that are not yet popular and are expensive. Therefore, age and the monthly disposable amount will also affect consumers’ purchase intention. Lastly, since the analysis process of this study only focused on the impact of perception of green value and animal welfare on the subjects’ attitudes and behaviors of the explained subjects, and the cumulative variance was 83.562%, it is speculated that plant-based meat alternatives products may have other values (such as environmental protection value, moral value, and various variables perceived value) which have not been explored.

Based on these limitations, this study suggested that future researchers should expand the research subjects to include all age groups or monthly disposable amount with experience in purchasing plant-based meat alternatives, incorporating variables such as moral value environmental value, and experiential value into the exploration, and conduct comparative measurements between age groups and monthly disposable amount through post hoc tests to find the actual main customer groups of plant-based meat alternatives and make the research results more complete.

## Figures and Tables

**Figure 1 foods-11-02423-f001:**
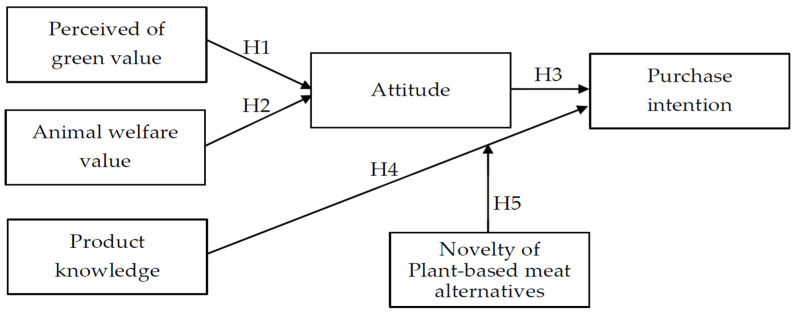
Research framework.

**Figure 2 foods-11-02423-f002:**
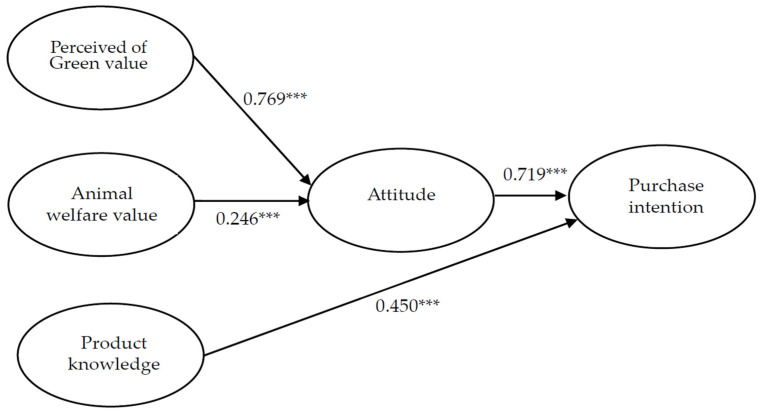
Results of structural equation modeling Note. *** *p* < 0.001.

**Figure 3 foods-11-02423-f003:**
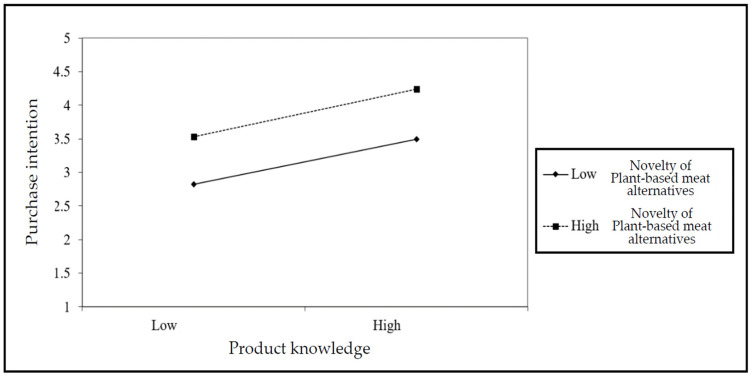
The interactive effect of product knowledge, purchase intention, and Novelty of Plant-based meat alternatives.

**Table 1 foods-11-02423-t001:** Sample characteristics.

*N* = 376	Item	*N*	Percentage
Gender	Male	103	27.4%
	Female	273	72.6%
Monthly disposable amount	Less than NTD 4999	82	21.8%
	Between NTD 5000 and NTD 9999	183	48.7%
	Between NTD 10,000 and NTD 14,999	75	19.9%
	Between NTD 15,000 and NTD 19,999	11	2.9%
	Less than NTD 20,000	25	6.6%
Ever purchased plant meat	Yes	122	32.4%
	No	254	67.6%
Pay attention to environmental issues	Yes	317	84.3%
	No	59	15.7%

Note. NTD: New Taiwan dollars.

**Table 2 foods-11-02423-t002:** Constructs/variables and corresponding measuring statements are included in the questionnaire.

Construct/Variable	Number of Statements	Measuring Items	Sources of Adoption
Perceived of green value	4	I think plant meat is an environmentally friendly product. I believe that purchasing plant-based meat products is helpful for environmental sustainability (e.g., reducing carbon emissions, reducing resource consumption). I think the performance of plant-based meat in promoting environmental sustainability meets my expectations. I think plant meat is better for the environment than other meats (e.g., beef, pork).	Williams& Soutar [74]
Animal welfare value	3	I don’t think eating meat from livestock raised animals is in line with animal welfare, because their lives deserve to be respected. I think consumers should stop killing animals, even if it costs some people their jobs. I think it is morally wrong to kill animals for the sake of appetite.	Zhung & Liu [75]
Attitude	2	I think it is right to buy plant-based meat products for environmental sustainability. Buying plant-based meat products for environmental sustainability makes me feel good.	Tsen et al. [76]
Purchase intention	3	I would like to buy plant-based meat that promotes environmental sustainability. I would like to recommend family and friends to buy plant—based meat that promotes environmental sustainability. I am willing to continue to buy plant-based meat that promotes environmental sustainability.	Follows & Jobber [77]
Product knowledge	4	Information about plant-based meat products influences my intention to purchase. For environmental sustainability, I will take the initiative to learn about plant-based meat products. If I understand the differences between various plant-based meat products (e.g.,: vegan, green certification, etc.), it may affect my choice when buying plant-based meat products. If I know about plant-based meat products, I will be willing to share with friends and family.	Brucks [78]

**Table 3 foods-11-02423-t003:** Results of factor loading, reliability, and validity.

Items	Factor Loading	Cronbach’s α	CR	AVE
Perceived of green value		0.868	0.874	0.634
1. I think plant meat is an environmentally friendly product.	0.85			
2. I believe that purchasing plant-based meat products is helpful for environmental sustainability (e.g., reducing carbon emissions, reducing resource consumption).	0.88			
3. I think the performance of plant-based meat in promoting environmental sustainability meets my expectations.	0.74			
4. I think plant meat is better for the environment than other meats (e.g., beef, pork).	0.71			
Animal welfare value		0.870	0.869	0.690
1. I don’t think eating meat from livestock raised animals is in line with animal welfare, because their lives deserve to be respected.	0.84			
2. I think consumers should stop killing animals, even if it costs some people their jobs.	0.86			
3. I think it is morally wrong to kill animals for the sake of appetite.	0.79			
Attitude		0.776	0.781	0.640
1. I think it is right to buy plant-based meat products for environmental sustainability.	0.82			
2. Buying plant-based meat products for environmental sustainability makes me feel good.	0.78			
Purchase intention		0.910	0.912	0.775
1. I would like to buy plant-based meat that promotes environmental sustainability.	0.92			
2. I would like to recommend family and friends to buy plant-based meat that promotes environmental sustainability.	0.84			
3. I am willing to continue to buy plant-based meat that promotes environmental sustainability.	0.88			
Product knowledge		0.788	0.788	0.487
1. Information about plant-based meat products influences my intention to purchase.	0.53			
2. For environmental sustainability, I will take the initiative to learn about plant-based meat products.	0.77			
3. If I understand the differences between various plant-based meat products (e.g., vegan, green certification, etc.), it may affect my choice when buying plant-based meat products.	0.66			
4. If I know about plant-based meat products, I will be willing to share with friends and family.	0.80			
Novelty of plant-based meat alternatives		0.879	0.879	0.784
1. I like to try the novelty of plant-based meat products.	0.91			
2. I would like to try novelty of plant-based meat products.	0.86			

Note: CR: Composite reliability; AVE: Average variance extracted.

**Table 4 foods-11-02423-t004:** Means, standard deviations, and correlations of constructs.

Construct	Mean	S.D.	1	2	3	4	5	6	7	8	9	10
1. Gender	1.730	0.447	1									
2. Monthly disposable amount	2.240	1.039	−0.203 **	1								
3. Ever purchased plant meat	1.680	0.469	−0.018	−0.092	1							
4. Pay attention to environmental issues	1.160	0.364	−0.014	0.020	0.096	1						
5. Perceived of green value	4.044	0.729	0.010	−0.061	−0.126 *	−0.194 **	1					
6. Animal welfare value	2.896	1.035	0.094	−0.058	−0.212 **	−0.197 **	0.344 **	1				
7. Attitude	3.735	0.811	0.028	−0.031	−0.170 **	−0.197 **	0.591 **	0.458 **	1			
8. Purchase intention	3.528	0.919	−0.006	−0.024	−0.278* *	−0.283 **	0.551 **	0.520 **	0.699 **	1		
9. Product knowledge	3.817	0.703	0.042	0.063	−0.162 **	−0.333 **	0.528 **	0.387 **	0.542 **	0.571 **	1	
10. Novelty of Plant-based meat alternatives	3.743	0.935	−0.086	0.032	−0.245 **	−0.191 **	0.440 **	0.301 **	0.471 **	0.580 **	0.501 **	1

Note: *N* = 376; * *p* < 0.05.; ** *p* < 0.01.

**Table 5 foods-11-02423-t005:** The model’s standardized regression weights, *t*-values, and hypothesis.

Path	Standardized Regression Weight	*t*-Value	Hypothesis	Verification
Directed effect of the integrative model				
Step 1: Independent variable—Product knowledge	0.571 ***	0.374 ***	0.374 ***	
Perceived of green value → Attitude (γ_11_)	0.769	10.676 ***	H1 *	Supported
Animal welfare value → Attitude (γ_12_)	0.246	6.148 ***	H2 *	Supported
Attitude → Purchase intention (β_12_)	0.719	9.513 ***	H3 *	Supported
Product knowledge → Purchase intention (γ_23_)	0.450	6.375 ***	H4 *	Supported
χ^2^/df = 2.351, GFI = 0.930, AGFI = 0.900, CFI = 0.964, NFI = 0.940, SRMR = 0.049, RMSEA = 0.060				

Note: *t* > 3.29, *** *p* < 0.001; * indicates the hypothesis was supported; GFI: Goodness of fit index; AGFI: Adjusted goodness of fit index; CFI: Comparative fit index; NFI: Normed-fit index; RMR: Standardized root mean square residual; RMSEA: Root mean square error of approximation.

**Table 6 foods-11-02423-t006:** Results of hierarchical regression analysis.

Variables	Purchase Intention		
	Model 1	Model 2	Model 3
Step 1: Independent variable—Product knowledge	0.571 ***	0.374 ***	0.374 ***
Step 2: Moderator—Novelty of Plant-based meat alternatives		0.393 ***	0.396 ***
Step 3: Interaction—Product knowledge x Novelty of plant-based meat alternatives			0.014
R^2^	0.325	0.441	0.441
ΔR^2^		0.116	0.000
F	180.476 ***	147.157 ***	97.914 ***

Note. *** *p* < 0.001.

## Data Availability

Data is contained within the article.

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
