# Peer review of "The Effect of Novel and Environmentally Friendly Foods on Consumer Attitude and Behavior: A Value-Attitude-Behavioral Model"

_foods, 2022, doi:10.3390/foods11162423_

Round 1
Reviewer 1 Report
The submitted paper is well written.
-I leave the authors just some suggestions:
Although if the sample analyzed is a convenience sample the authors should emphassize that the used sample (university students) deserves attention for the investigated questions:
I suggest the author to evidence several aspects widely analized on tpurchasing and consumption habits. They could add references regarding the links between meat consumption /gender/household composition/animal welfare in sections:
- 2.3.1. Perception of Green Value and Attitude
- 2.3.2. Animal Welfare Value and Attitude
Many recent papers focus on those topics. As example I report:
When citing the Thory of plannec behaviour it could be useful to refer to references such as:
Lai, Amanda Elizabeth, et al. "Two sides of the same coin: Environmental and health concern pathways toward meat consumption." Frontiers in Psychology 11 (2020): 578582.
Tales 3 and 6 must be reformatted .. in the present version of the paper they are not readable.
Author Response
Revision Response to Reviewer’s Comments (Reviewer 1)
Thank you very much for your insightful comments and suggestions. We believe your comments and suggestions are appropriate and valuable to improve the manuscript's quality considerably. We have revised our paper in light of your comments and instructions.
The point-by-point concerns of the reviewers are discussed as follows:
|
Comments and Suggestions for Authors Overall, the comments have been well addressed. |
|
|
1 |
The submitted paper is well written. -I leave the authors just some suggestions: |
|
Revision Feedbacks |
Thank you for your encouragement, and we will try to improve it. |
|
2 |
Although if the sample analyzed is a convenience sample, the authors should emphasize that the used sample (university students) deserves attention for the investigated questions: I suggest the author to evidence several aspects widely analyzed on purchasing and consumption habits. They could add references regarding the links between meat consumption /gender/household composition/animal welfare in sections: - 2.3.1. Perception of Green Value and Attitude - 2.3.2. Animal Welfare Value and Attitude Many recent papers focus on those topics. As example I report: When citing the Theory of planned behavior, it could be useful to refer to references such as: Lai, Amanda Elizabeth, et al. "Two sides of the same coin: Environmental and health concern pathways toward meat consumption." Frontiers in Psychology 11 (2020): 578582. |
|
Revision Feedbacks |
Thank you for your kind reminder and sound advice. We have revised it as follows: Page 4; lines 19-24 2.3.1. Perception of Green Value and Attitude In terms of consumers' attitudes towards plant-based meat alternatives, the belief that people surrounding the individual are reducing their meat consumption should activate their moral obligation to follow in the same direction [45]. Furthermore, research shows that the stronger the perception that significant others approve of one’s reduction in meat consumption, the greater the individual’s awareness of the consequences of this behavior [46]. Furthermore, people care whether their behavior is moral to others and are motivated to maintain a positive moral self-image [47] and to belong to a moral group [48]. 45. Lai, A. E., Tirotto, F. A., Pagliaro, S., & Fornara, F. (2020). Two Sides of the. Same Coin: Environmental and. Health Concern Pathways Toward. Meat Consumption. Front. Psychol., 11, 578582. https://doi.org/10.3389/fpsyg.2020.578582 46. Fornara, F., Pattitoni, P., Mura, M., & Strazzera, E. (2016). Predicting intention to improve household energy efficiency: the role of value-belief-norm theory, normative and informational influence, and specific attitude. J. Environ. Psychol., 45, 1-10. https://doi.org/10.1016/j.jenvp.2015.11.001 47. Jordan, A. H., & Monin, B. (2008). From sucker to saint: moralization in response to self-threat. Psychol. Sci., 19(8), 809-815. https://doi.org/10.1111/j.1467-9280.2008.02161.x 48. Ellemers, N., & van den Bos, K. (2012). Morality in groups: on the social- regulatory functions of right and wrong. Soc. Personal. Psychol. Compass., 6(12), 878–889. https://doi.org/10.1111/spc3.12001
|
|
Revision Feedbacks |
Page 4; lines 39-44 2.3.2. Animal Welfare Value and Attitude Since animal welfare is a moral issue [51], people's attitudes towards animal welfare can vary depending on their culture, area of residence, time, or personal factors [52-53]. Studies in the related literature pointed out that vegetarians based on animal welfare believe that there is a solid emotional bond between humans and animals [54]. Some people have an attitude of rejecting animal products based on animal welfare or health factors [55] and believe that animals have the same perceptive capacity as consumers and should be treated with animal welfare value [56]. Based on those mentioned above, people are motivated to reduce their meat consumption for different reasons, for example, animal welfare, environmental and health concerns [45; 57]. These motivations are not mutually exclusive; however, it is possible to identify a trend where animal-rights and ecological concerns are more likely to be found in those who completely exclude meat from their diet, whereas less morally relevant reasons, such as health concerns, seem to mainly motivate those who deliberately choose only to reduce meat consumption [58-59]. 45. Lai, A. E., Tirotto, F. A., Pagliaro, S., & Fornara, F. (2020). Two Sides of the. Same Coin: Environmental and. Health Concern Pathways Toward. Meat Consumption. Front. Psychol., 11, 578582. https://doi.org/10.3389/fpsyg.2020.578582 57. Sanchez-Sabate, R., & Sabaté, J. (2019). Consumer attitudes towards environmental concerns of meat consumption: a systematic review. Int. J. Environ. Res. Public Health, 16(7),1220. https://doi.org/10.3390/ijerph16071220 58. De Backer, C. J., & Hudders, L. (2015). Meat morals: relationship between meat consumption consumer attitudes towards human and animal welfare and moral behavior. Meat Sci., 99, 68–74. https://doi.org/10.1016/j.meatsci.2014.08.011 59. Rosenfeld, D. L., Rothgerber, H., & Tomiyama, A. J. (2020). From mostly vegetarian to fully vegetarian: meat avoidance and the expression of social identity. Food Qual. Prefer. 85, 103963. https://doi.org/10.1016/j.foodqual.2020.103963 60. |
|
3 |
Tales 3 and 6 must be reformatted .. in the present version of the paper, they are not readable. |
|
Revision Feedbacks |
Thanks for your comments, and we have revised it. Page 9; Table3
|
|
Revision Feedbacks |
Page 12; Table6 |

Reviewer 2 Report
Overall this seems to be a sound and well-articulated manuscript based on the VAB model.
The sampling method leads to major concerns about the validity, applicability, and generalizability of the findings. Given the severe demographic skewness, it would seem essential (and rather easy) to use sampling weights (post-stratification) and re-estimate the model to provide the basis for a robustness check. Short of that or a similar procedure, the findings will be picked apart by methodologically sensitive readers and the broader impact of the findings will be severely attenuated.
A less impactful, but still important, matter is the reference to "hierarchical" modeling. The stage-wise procedure discussed in the manuscript is far different from the contemporary use of "hierarchical regression" to mean multilevel modeling or the application of mixed linear models for data measured on both individual and aggregate levels. A related concern is that stage-wise models are open to criticism that the results may be capitalizing on chance and not reflect particularly meaningful findings. A Bonferroni correction or false discovery rate adjustment could be helpful.
Author Response
Revision Response to Reviewer’s Comments (Reviewer 2)
Thank you very much for your insightful comments and suggestions. We believe your comments and suggestions are appropriate and valuable to improve the manuscript's quality considerably. We have revised our paper in light of your comments and instructions.
The point-by-point concerns of the reviewers are discussed as follows:
|
Comments and Suggestions for Authors Overall, the comments have been well addressed. |
|||||||||||||||||||||
|
1 |
Overall this seems to be a sound and well-articulated manuscript based on the VAB model. |
||||||||||||||||||||
|
Revision Feedbacks |
Thank you for your approval and comments. |
||||||||||||||||||||
|
2 |
The sampling method leads to major concerns about the validity, applicability, and generalizability of the findings. Given the severe demographic skewness, it would seem essential (and rather easy) to use sampling weights (post-stratification) and re-estimate the model to provide the basis for a robustness check. Short of that or a similar procedure, the findings will be picked apart by methodologically sensitive readers and the broader impact of the findings will be severely attenuated.
|
||||||||||||||||||||
|
Revision Feedbacks |
Thank you so much for your kind suggestions. This is very helpful in clarifying the issue of gender ratio in our sample. We have made some efforts to revise it. Please see the following sections added in 3.1. Sample and Data Collection and 6.2. Limitations. Thank you again!
For Taiwanese consumers, the novelty of plant-based meat alternatives is not a familiar novelty to university students, so this study tried to choose the department of food science or nutrition of Taiwan universities as the research sample. According to data from the Taiwan Ministry of Education for the 2022-2022 school year, the ratio of male to female students in food science or nutrition is 25-28% male and 72-75% female, which is similar to the percentage of 27.4% to 72.6% in the research sample of this study. For the rigor of the study, the chi-square test was also used to determine the representativeness of the recovery. The results showed that when we calculated the ratio of 30% and 70% or 25% and 75% of the natural mother group, the ratio of men and female recovered in this study was 27.4% and 72.6%. The calculated sample proportions were not significantly different. Therefore, we know that the ratio of males to females in the samples recovered in this study is the same as that of the parent group of students in Taiwan’s universities' food science or nutrition department. Sun Yat-Sen Medical University Male to Female Ratio
|
||||||||||||||||||||
|
Revision Feedbacks |
China Medical University Male to Female Ratio
National Chung Hsing University Male to Female Ratio
|
||||||||||||||||||||
|
|
Page 6 3.1. Sample and Data Collection The research participants in this study were students in the department of food science or nutrition of Taiwan University. This study used a paper questionnaire and adopted purposive sampling to conduct a research investigation.
Page 15-16 6.2. Limitations For Taiwanese consumers, the novelty of plant-based meat alternatives is not familiar to university students, so this study tried to choose the department of food science or nutrition of Taiwan universities as the research sample. According to data from the Taiwan Ministry of Education for the 2022-2022 school year, the ratio of male to female students in food science or nutrition is 25-28% male and 72-75% female, which is similar to the percentage of 27.4% to 72.6% in the research sample of this study. Only students in the Department of Food Science or nutrition at Taiwan University were taken as research participants, resulting in a limited sampling scope so that the results of this study cannot be extrapolated to consumers of all age groups. |
||||||||||||||||||||
|
3 |
A less impactful, but still important, matter is the reference to "hierarchical" modeling. The stage-wise procedure discussed in the manuscript is far different from the contemporary use of "hierarchical regression" to mean multilevel modeling or the application of mixed linear models for data measured on both individual and aggregate levels. A related concern is that stage-wise models are open to criticism that the results may be capitalizing on chance and not reflect particularly meaningful findings. A Bonferroni correction or false discovery rate adjustment could be helpful. |
||||||||||||||||||||
|
Revision Feedbacks |
Thanks for your kind reminder and for suggesting post hoc testing using Bonferroni correction to mitigate the problem of multiple testing. Bonferroni correction can compensate for over-expanded type-one error probability with corrected significance levels.
This research focuses on a single level of analysis and does not explore multilevel issues. Furthermore, the five research hypotheses in the research framework, H1-H4, use AMOS for structural equation modeling (SEM) analysis, and H5 adopts hierarchical regression analysis of SPSS analysis.
Page 11-12 4.3. Testing Interference effects For research rigor, we also used SPSS PROCESS model 1 to process the H5 moderating effect to verify whether the analysis results were the same as the results of hierarchical regression analysis using SPSS.
This study uses product knowledge as an independent variable, purchase intention as a dependent variable, and novelty of plant-based meat alternatives as a moderator, using PROCESS model 1 (for simple moderation) with bootstrapping, the number of bootstrap samples is set to 5,000 to examine the relationship between the novelty of plant-based meat alternatives to product knowledge and purchase intention.
The confidence interval between the low confidence interval (LLCI) and the high confidence interval (ULCI) was found to have a value of 0, so there was no moderating effect (see the figure below for details). It is the same as the actual result of using SPSS's hierarchical regression analysis to regulate the effect of H5. Testing moderating effect by SPSS PROCESS Model 1 |
||||||||||||||||||||

Round 2
Reviewer 2 Report
The authors do not seem to have integrated all relevant responses into the document, but I believe what they have done is sufficient to move the manuscript along to publication.